# Regression for the Mean: Auto-Evaluation and Inference with Few Labels through Post-hoc Regression

Benjamin Eyre [1] [*]    David Madras [2]

## Abstract

The availability of machine learning systems that can effectively perform arbitrary tasks has led to synthetic labels from these systems being used in applications of statistical inference, such as data analysis or model evaluation. The Prediction Powered Inference (PPI) framework provides a way of leveraging both a large pool of pseudo-labelled data and a small sample with real, high-quality labels to produce a low-variance, unbiased estimate of the quantity being evaluated for. Most work on PPI considers a relatively sizable set of labelled samples, which can be resource intensive to obtain. However, we find that when labelled data is scarce, the PPI++ method can perform even worse than classical inference. We analyze this phenomenon by relating PPI++ to ordinary least squares regression, which also experiences high variance with small sample sizes, and use this regression framework to better understand the efficacy of PPI. Motivated by this, we present two new PPI-based techniques that leverage robust regressors to produce even lower variance estimators in the few-label regime.

## 1. Introduction

The deployment of machine learning (ML) systems into several areas of modern life has the potential to lead to several positive outcomes, such as faster and more precise medical care (Rajpurkar et al., 2017), improved educational tools (Jaiswal & Arun, 2021; Lee, 2024), and a wide variety of professional assistive technologies (Dehaerne et al., 2022). However, the ever increasing integration of these systems has resulted in a more drastic need for effective evaluation techniques for determining when these systems suffer from

systematic errors (DeGrave et al., 2021). Accurate evaluation in the era of large language models (LLMs) presents a scaling challenge. Traditional evaluation schemes involve collecting several annotated samples from human labellers, and with ML systems quickly changing (Tu et al., 2023), the repeated collection of this data can quickly exhaust time and resources.

Recent advances in the development of LLMs have resulted in the widespread availability of reasonably strong predictive models for arbitrary tasks. The proliferation of these models has made the practice of replacing human-annotated labels with the outputs of an LLM for the sake of evaluation, or automatic evaluation, more feasible. Inherent bias in these LLM predictions leads to potentially inaccurate evaluations, even in the case where many examples are available (Angelopoulos et al., 2023a). Proposed frameworks such as Prediction Powered Inference (PPI) (Angelopoulos et al., 2023a) offer a way to remove the statistical bias of these predictions with a small pool of labelled data.

While previous works studying the PPI framework have focussed on circumstances where 50 or more (usually 200+) labelled examples are available (Boyeau et al., 2024; Angelopoulos et al., 2023b; Zrnic & Candès, 2024b;a), the applicability of PPI in the few-label regime has not been thoroughly investigated — in this work, we find that empirically, PPI performs poorly (sometimes worse than classical inference) when very few labelled examples are available. We argue this is an important case, as many developers of ML systems will rarely have access to a large evaluation set for every criteria they wish to evaluate their system for, e.g. when developing generative models, important evaluations are often highly qualitative and potentially time-consuming. In these cases, developers may wish to use a small pool of hand-labelled examples as a way to guide design decisions.

With this in mind, ensuring evaluation with few labels is as efficient and accurate as possible is important for the development of reliable ML systems. Specifically in the case of generative model evaluation, the PPI framework is a natural candidate, as the large set of unlabelled data PPI requires can be directly generated by the model itself. Efficient inference with few labels is also relevant to research in the sciences, where collecting more data may be costly but

---

[*] Work partially completed while this author was at Google DeepMind. [1]Columbia University [2]Google DeepMind. Correspondence to: Benjamin Eyre <beneyre@cs.columbia.edu>.

*Proceedings of the $42^{nd}$ International Conference on Machine Learning*, Vancouver, Canada. PMLR 267, 2025. Copyright 2025 by the author(s).

annotating unlabelled data may be less resource intensive. In this paper, we work towards the goal of improving few-label auto-evaluation by proposing modifications to the PPI framework which can produce lower variance estimates in the low-label regime. Specifically, we do this through emphasizing and re-interpreting the critical role of the tuning parameter $\lambda$. Our contributions are:

1. Providing a theoretical analysis of the PPI++ method in the context of mean estimation, and how it relates to univariate regression.

2. Positing two new extensions for PPI inspired by univariate regression techniques that improve upon the base method.

3. Applying these techniques to the task of *feature generation rate estimation* in research and LLM evaluation.

## 2. Background

In this section, we provide details on the estimation problems we approach, their associated data generative process, and its relationship with contemporary automatic evaluation schemes.

### 2.1. Estimating Feature Generation Rate

We are interested in the properties of some outputs $X$ generated from some distribution P, $X \sim P$. These outputs can consist of any modality (e.g. text, images, video) and the distribution can take any form (e.g. a complex generative model). Consider the binary function $h(X) \in \{0, 1\}$ that outputs 1 if $X$ presents a certain feature of interest, and 0 if X does not present that feature. $h(X)$ can represent quantitative features such as the presence of a word in text or the value of a certain pixel in an image, or more subjective features such as toxicity in text. This notion can be extended to non-binary functions, but this work will focus on the binary case. We wish to estimate the quantity $\mathbb{E}_P[h(X)]$. The best known estimator is the *classical* estimator, which is simply the sample mean from a finite sample of $n$ example, label pairs $\mathcal{D}_n = \{(X_i, h(X_i))\}_{i=1}^n$:

$$\mu_{h,P} := \mathbb{E}_P[h(X)], \quad \hat{\mu}_h := \frac{1}{n}\sum_{i=1}^n h(X_i) \qquad (1)$$

This estimate is unbiased, and its variance will decay in order $\frac{1}{n}$. However, in the regime where labels are scarce or expensive to procure the variance on the sample mean will be high, leading to low-fidelity estimates of the mean.

### 2.2. Prediction Powered Inference for Mean Estimation

We next examine how a strong predictive model could be used to complete this estimation task. In our set up, we can consider another binary function, $f(X) \in \{0, 1\}$ that is meant to approximate $h(X)$. We can estimate $\mu_{h,P}$ by instead taking the sample mean of a finite pseudolabelled sample $\mathcal{D}_N = \{(X_i^u, f(X_i^u))\}_{i=1}^N$. Seeing as unlabelled data $X$ is often plentiful and $f(X)$ can be generated without human intervention, one can efficiently reach a value of $N$ where the variance of the sample mean is minimized. However, $f(X)$ may be biased. More rigorously:

$$\mu_{f,P} := \mathbb{E}_P[f(X)], \quad |\mu_{f,P} - \mu_{h,P}| > 0. \qquad (2)$$

If this is true, regardless of how large our sample of $N$ examples is, estimating the mean using pseudolabels will have an irreducible amount of error.

In response to the shortcomings of this automatic evaluation approach, Angelopoulos et al. (2023a) leverage a small pool of in-distribution labelled examples along with a larger sample of pseudolabelled examples to create a low variance unbiased estimator in the Prediction Powered Inference (PPI) framework. The PPI estimate takes as input both the labelled dataset $\mathcal{D}_n$ and the pseudolabelled dataset $\mathcal{D}_N$, as well as a tuning parameter $\lambda \in \mathbb{R}$, and estimates $\mu_{h,P}$ as:

$$\hat{\mu}_{PPI} := \frac{1}{N}\sum_{i=1}^N \lambda f(X_i^u) + \frac{1}{n}\left(\sum_{i=1}^n h(X_i) - \lambda f(X_i)\right). \quad (3)$$

This approach, as well as the methods that have succeeded it (Angelopoulos et al., 2023b; Zrnic & Candès, 2024a; Boyeau et al., 2024), makes use of both the strong statistical power that traditional automatic evaluation schemes promise, as well as the asymptotic benefits of being unbiased provided by traditional evaluation.

## 3. Related Work

Since its inception, the PPI framework has been the subject of much investigation, both in terms of application and improving the method. Zrnic & Candès (2024a) propose a technique for determining which samples within a batch of unlabelled data to collect labels for to achieve the best possible performance with PPI. Zrnic & Candès (2024b) show how one can use cross-fitting to be able to use PPI even when a pre-trained predictive model is unavailable. Fisch et al. (2024) demonstrate that using subgroup information can provably improve the estimates from PPI. While these works focus on how to apply the PPI framework under different assumptions, we instead focus on improving the efficacy of the method in its original setting. Boyeau et al. (2024) were the first to propose how the PPI framework can be used to improve contemporary automatic evaluation schemes. We build upon this work by investigating new use cases and improving performance in the low-label regime.

Several techniques for supplementing a model's training data with synthetic data have been proposed. This includes both data augmentation techniques such as Mixup (Zhang, 2017) and those used in reinforcement learning (Pitis et al., 2022; Osiński et al., 2020). Improved generative models have provided a new avenue for researchers and practitioners alike to create new datasets for both training and evaluation (Neuhausen et al., 2020). We contribute to this branch of research by building upon a method for evaluating a generative model with synthetic data generated by the model itself.

The use of an additional, correlated variable as a way to reduce the variance of an estimator is a well studied technique in the fields of statistics and machine learning. Previous works have specifically investigated how one can reduce the variance in mean estimation using a control variate constructed with unlabelled data in a regime known as semi-supervised learning (Zhang et al., 2019; Zhang & Bradic, 2022). Furthermore, the use of different regression methods and control variates has also been explored (South et al., 2023; Blanchet et al., 2024). In this work, we use this existing literature to better understand and improve the PPI++ method.

## 4. Using Regression for Improving PPI

In this section, we discuss our approach to improving the PPI estimator in the few-label regime. Our approach prioritizes reducing variance and focuses on the role of $\lambda$ in PPI, re-interpreting the task of choosing $\lambda$ as post-hoc regression. We first draw the connection between optimizing $\lambda$ and linear regression, and then discuss how choosing $\lambda$ correctly can be challenging in the low-label regime, resulting in high variance estimates. We then highlight two tools from the regression literature which can be used to improve these estimates, and demonstrate how they fit into the PPI framework.

### 4.1. Optimal $\lambda$ and the Regression Coefficient

To understand how selecting $\lambda$ appropriately can reduce the variance of $\hat{\mu}_{PPI}$, we first provide an expression for the variance of $\hat{\mu}_{PPI}$, and decompose it into $Var[\mathcal{D}_n]$ and $Var[\mathcal{D}_N]$:

$$Var[\hat{\mu}_{PPI}] = \underbrace{Var[\frac{1}{N}\sum_{i=1}^{N}\lambda f(X_i^u)]}_{Var[\mathcal{D}_N]} + \underbrace{Var[\frac{1}{n}(\sum_{i=1}^{n}h(X_i) - \lambda f(X_i))]}_{Var[\mathcal{D}_n]}. \quad (4)$$

Note that we are able to distribute the variance operator this way as the two samples are i.i.d. Given that we assume that $n << N$ (e.g. we can generate or collect a large amount of unlabeled data cheaply), $Var[\mathcal{D}_N]$ will be negligible and we may turn our focus to $Var[\mathcal{D}_n]$. We can further decompose this variance as:

$$\begin{aligned} Var[\mathcal{D}_n] &= \frac{1}{n^2}\sum_{i=1}^{n}Var[h(X_i) - \lambda f(X_i)] \\ &= \frac{1}{n^2}\sum_{i=1}^{n}Var[h(X)] + \lambda^2 Var[f(X)] \\ &\quad - 2\lambda Cov[h(X), f(X)]. \end{aligned} \quad (5)$$

The first term in this sum is equal to the variance of the classical estimate (the labelled sample mean) for $\mu_{h,P}$. Therefore, $\hat{\mu}_{PPI}$ will have lower variance than the sample mean estimate whenever $\lambda^2 Var[f(X)] - 2\lambda Cov[h(X), f(X)] < 0$.

One can find a simple expression for the optimal $\lambda$ given $\mathcal{D}_n$ by simply taking the derivative of Equation 5 with respect to $\lambda$ and setting it to zero, which yields:

$$\lambda_{Opt} = \frac{Cov[h(X), f(X)]}{Var[f(X)]}. \quad (6)$$

Notably, Angelopoulos et al. (2023b) choose $\lambda$ to minimize both $Var[\mathcal{D}_n]$ and $Var[\mathcal{D}_N]$ and arrive at a similar expression[1]. The quantity expressed in Equation 6 is known as the regression coefficient (Kenney & Keeping, 1962), and is the solution to univariate ordinary least squares regression.

We emphasize this insight into the role of $\lambda$ as a *post-hoc regressor*. Previous works have encouraged an interpretation where $\lambda$ is intended to *interpolate* between two estimators: classical estimation at $\lambda = 0$ and "real PPI" at $\lambda = 1$. Indeed, Boyeau et al. (2024) describe $\lambda$ as a "tuning parameter ... in $[0, 1]$ ... When the synthetic data is very good, we can set $\lambda = 1$; when it is bad, setting $\lambda = 0$ will throw it away entirely." However, we suggest that $\lambda$ may more accurately be interpreted as a post-hoc transformation of $f$ to make it "closer to" $h$; that is, minimizing PPI variance through equivalently minimizing the mean-squared error of the post-hoc regression problem, which has the same form as Equation 5:

$$\begin{aligned} \frac{1}{n}\sum_{i=1}^{n}(h(X_i) - \lambda f(X_i) - b)^2 \approx \hat{Var}[h(X)] + \hat{Var}[\lambda f(X)] \\ - 2\hat{Cov}[h(X), \lambda f(X)]. \end{aligned} \quad (7)$$

---

[1]We note that when $n$ is small, this expression is unlikely to have converged in probability to a constant. Consequently, fitting $\lambda$ to the labelled data may result in $\hat{\mu}_{PPI}$ being a biased estimator in this small $n$ regime.

and admits the same minimizer (derivation details in Appendix B). Here, $b = \hat{\mu}_h - \hat{\mu}_{\lambda f}$ is the optimal ordinary least squares intercept coefficient, which is the difference between the sample means of the target $h(X)$ and transformed input $\lambda f(X)$. We note that this interpretation better aligns with the fact that PPI is still valid for $\lambda$ outside the $[0, 1]$ interval, and indeed in some situations may achieve its minimal variance at one of these values of $\lambda$[2].

## 4.2. Variance Reduction through Regularized Regression

Thinking of the $\lambda$-selection problem simply as univariate regression allows for new intuitions around the challenges of few-label PPI. It is known that ordinary least squares (OLS) is an unbiased estimator, but can suffer from high variance when few examples are available. In regression, a standard way to overcome issues with variance is to use *ridge regression*. This estimator imposes an L2 penalty in addition to the MSE to reduce the variance of the estimate (at the price of adding a small amount of bias). We propose using ridge regression to robustly estimate $\lambda$ in the case where $n$ is small. We specifically propose using the estimator:

$$\hat{\lambda}_\alpha := \frac{\hat{Cov}[h(X), f(X)]}{\hat{Var}[f(X)] + \alpha}. \tag{8}$$

This value yields the minimum mean squared error for the ridge regression problem with $\alpha \in \mathcal{R}$ as the ridge coefficient — $\alpha$ penalizes regression solutions with large L2-norm, having the effect of reducing the magnitude of the estimated $\lambda$. This estimated $\lambda$ is in turn used to calculate $\hat{\mu}_{PPI}$.

## 4.3. Variance Reduction through Non-linear Regression

Understanding the role of $\lambda$ as a regression coefficient can also lead us to investigate other hypothesis classes for regression. One may consider an arbitrary transform $g$ of $f(X)$, and it is clear to see that substituting $g(f(X))$ for $\lambda f(X)$ in Equation 5 can be done with no difficulties. Similarly, if one uses the intercept term $b = \hat{\mathbb{E}}[h(X)] - \hat{\mathbb{E}}[g(f(X))]$ and substitutes $g(f(X))$ for $\lambda f(X)$ in Equation 7, one finds that minimizing the mean squared error for an arbitrary function $g(X)$ is equivalent to minimizing the variance of $\hat{\mu}_{PPI}$.

What might be a useful $g$ to consider? In the rate estimation case (i.e. binary $h$), we note that it's somewhat unnatural to perform linear regression of $f \in [0, 1]$ onto $h \in \{0, 1\}$. Instead, we propose using the simple but more well-suited

function class of sigmoidal regressors for post-hoc regression of $f$, as well as an associated PPI estimate:

$$g(f(X)) := \frac{1}{1 + exp(-\alpha f(X) + \beta)}, \tag{9}$$

$$\hat{\mu}_{PPI_g} := \frac{1}{N} \sum_{i=1}^{N} g(f(X_i^u)) + \frac{1}{n}(\sum_{i=1}^{n} h(X_i) - g(f(X_i))). \tag{10}$$

where $\alpha, \beta$ are learned parameters. By using a function class which is better-suited to our problem domain, we hope to find useful transformations which are not possible with a linear regressor and thereby achieve greater variance reduction.

## 4.4. Discussion on Various Other Approaches in Existing Open Source Implementations

Angelopoulos et al. (2023b) demonstrate that when accounting for the added unlabelled term in Equation 3, the point estimate for the optimal lambda is a scaling of the regression coefficient dependant on the ratio of the size of the labelled and unlabelled datasets. We follow this methodology and scale both the PPI++ estimate $\hat{\lambda}_{Opt}$ and the Ridge-PPI estimate $\hat{\lambda}_\alpha$ by $(1 + \frac{n}{N})^{-1}$.

We additionally note that in a pre-existing open source implementation of the PPI++ algorithm[3], two additional heuristic inductive biases are used when calculating $\lambda$. First, a biased estimate of $Cov[h(X), f(x)]$ is used, where the sample covariance is divided by $n$ rather than $n - 1$. We use the unbiased estimate of the covariance in our implementations. Further, the open source implementation clips the estimated $\lambda$ to be within the interval $[0, 1]$. We do not clip our estimates of $\lambda$, as in principle the optimal regression coefficient may exist outside of that interval. We note that both of these heuristics may be helpful for improving the performance of PPI on some distributions as they both frequently produce smaller estimates of $\lambda$ — this is a similar reason to why we believe Ridge-PPI is helpful, which provides a principled approach to preventing overestimation of $\lambda$.

# 5. Experiments

In this section, we empirically analyze the performance of our proposed approaches in the few-label setting. We provide a description of the data that we analyze, the exact implementation of the methods that we use, and the results of our experiments.

---

[2]Angelopoulos et al. (2023b) note explicitly that in the mean estimation case (unlike more general cases), $\lambda$ need not be clipped at $[0, 1]$.

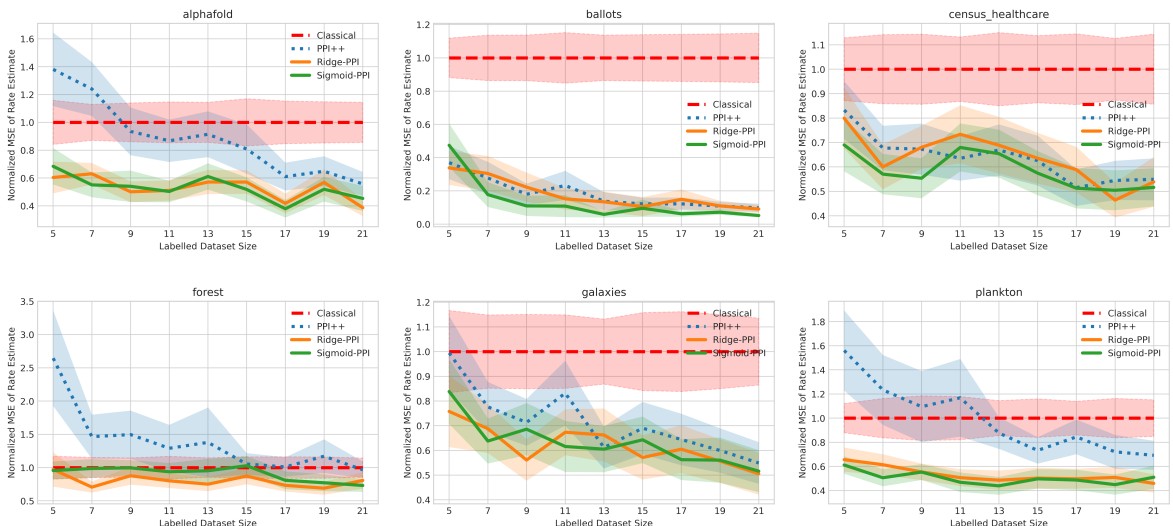

Figure 1: Estimation performance on research datasets. Average MAE of each estimate type is normalized by the average MAE of classical estimation at that batch size.

## 5.1. Datasets

### 5.1.1. RESEARCH DATA

We first apply our method to statistical inference for several real-world research questions. This suite of datasets, aggregated by Angelopoulos et al. (2023a), presents many instances where pseudolabelled data can be plentiful while gold standard labels are scarce. Some examples include using images of galaxies to estimate the occurrence rate of spiral galaxies and estimating the mean deforestation level in the Amazon over a specified time period. We defer to Angelopoulos et al. (2023a) for a full description of each of the datasets.

### 5.1.2. LLM REFUSAL DATA

For this section, we use a datatset of prompts and LLM outputs, along with human-made annotations indicating whether the output was an instance of *refusal*. LLMs can refuse to answer questions for a variety of reasons, such as when the model is prompted to complete a task outside its capabilities (Xu et al., 2024), asked to provide private information (Liu et al., 2024), or asked an inappropriate or unsafe question (Yuan et al., 2024). The dataset consists of over 50,000 prompt-answer pairs. Prompts consist of a wide variety of requests concerning several different topics, and the answers are sourced from multiple different publicly available LLMs. For our purposes, we will only use this data to understand how often a given LLM refuses to answer its prompt, rather than investigating the kinds of prompts the model refuses to answer. Given the high diversity of

---

prompts in the dataset, one cannot make conclusions regarding a model's sensitivity to any one specific subject matter using the bulk refusal rate, nor is one rate of refusal qualitatively better than any other. Instead, this dataset is intended to provide a relevant setting where qualitative annotations are required to evaluate a complex model behavior.

Within our previously described framework, examples $X_i$ are each a tuple containing a prompt and an answer from an LLM. Here, the labelling function $h(X)$ is equal to one when $X$ is an instance of refusal, and zero otherwise.

## 5.2. Experiments and Estimation Methods

To benchmark the efficacy of different estimation methods for the refusal rate, we conduct an experiment where we randomly sample both $\mathcal{D}_n$ and $\mathcal{D}_N$ from the larger pool of samples. We ablate over several different values of $n$ to compare each method's performance with different amounts of labelled data. We keep the amount of unlabelled data fixed at $N = 1000$. We measure performance of each method as the mean absolute error between the estimate and the true refusal rate over the entire dataset. For each setting of $n$, we sample $\mathcal{D}_n$ and $\mathcal{D}_N$ 350 times and compute the MAE over all trials.

In our experiments with the LLM Refusal Dataset, we experiment with four different methods for estimating the refusal rate:

- **Classical**: Classical estimation using the sample mean from the labelled data $\mathcal{D}_n$.

- **PPI++**: Estimating the refusal rate using both $\mathcal{D}_n$ and

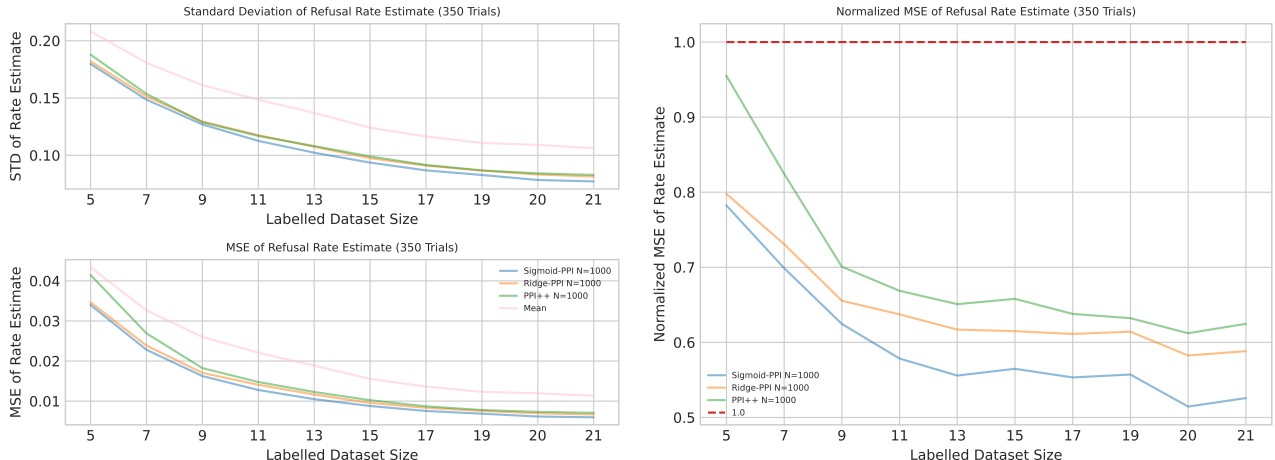

(a) Standard deviation of estimates (top) and average MAE of esti-(b) Average MAE of different estimates normalized by the average mates (bottom) for different estimation methods.    MAE of classical estimation at that batch size.

Figure 2: Estimation performance on the LLM Refusal Dataset.

$\mathcal{D}_N$ through $\hat{\mu}_{PPI}$. Here, we use the sample estimate of $\lambda_{Opt}$ based off Equation 6.

- **Ridge-PPI** (Ours): Similar to PPI++, but we instead fit $\lambda$ using ridge regression as seen in Equation 8. Here, we perform cross validation over the labelled set to determine the ridge parameter $\alpha$.

- **Sigmoid-PPI** (Ours): We follow the procedure described in Section 4.3 to fit a sigmoidal function on our labelled data from $f(X)$ to $h(X)$ and then use that to make a PPI estimate. Here we also use L2 regularization and choose the regularization parameter based on cross validation, similar to Ridge-PPI.

### 5.3. Results

#### 5.3.1. RESEARCH DATA

We find that both Ridge-PPI and Sigmoid-PPI are able to either match or exceed the performance of both classical estimation and PPI++ across each of the settings (Fig. 1). This includes the plankton, alphafold, and forest datasets, where traditional PPI++ fails to outperform classical for several settings of $n$ while our methods consistently match or exceed this performance. While across several of the datasets Ridge-PPI and Sigmoid-PPI are able to cut the error of the leading baseline by a significant fraction, other datasets such as ballots, forest, and census_healthcare show a more modest improvement. In sections 5.4 and 6.1, we further investigate this phenomenon and present hypotheses on the conditions that lead to it.

#### 5.3.2. LLM REFUSAL DATA

While each of the PPI based methods are able to achieve superior MAE over classical estimation, we find that both Ridge-PPI and Sigmoid-PPI are able to improve upon the performance of PPI++ for all small values of $n$ (Figure 2, left). We also observe that the standard deviation of these estimates is smaller, indicating that these techniques have indeed reduced the variance. This effect can most prominently be seen for smaller labelled sets (Figure 2, right), where we see that our improved PPI methods can reduce the MAE of classical estimation by over a quarter.

### 5.4. Distributional Influence on PPI's Efficacy

The underlying data distribution has the potential to effect the efficacy of each estimation method. To study how the variance of the target variable $h(X)$, the predictions $f(X)$, and the covariance between these variables effect performance, we perform the experiments described in Section 5.2 for each individual LLM represented in the LLM Refusal Dataset.

We find that the efficacy of PPI++ in comparison to classical estimation can vary highly across different distributions of samples. In some distributions, PPI++ cuts the error of classical estimation by roughly 30%, with our methods producing similar gains (Figure 3, right). Under other distributions, PPI++ performs worse than classical estimation; PPI++ can accrue 20% more error than classical estimation when applied to certain distributions (Figure 3, left). However, in these circumstances, we find that both Ridge-PPI and Sigmoid-PPI perform 10% better than classical estimation, demonstrating a circumstance in which our methods

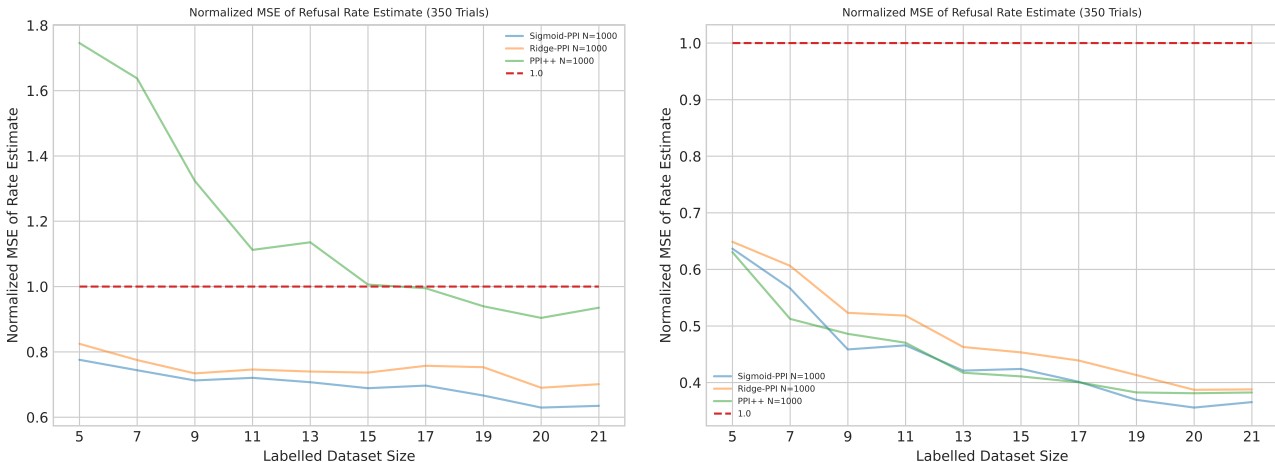

Figure 3: Estimation performance on two different subsets of the LLM refusal dataset normalized by the average MAE of classical estimation at that batch size. We note that $Var[f(X)]$ is twice as large on the right (where PPI++ succeeds) as on the left (where it fails).

are able to overcome the shortcomings of PPI++.

We suspect that the poor performance of PPI++ on distributions like this one may be related to low variance of $f(X)$. Since $Var[f(X)]$ appears in the denominator of Equation 6, potential errors from overestimating $Cov[f(X), h(X)]$ are amplified when $Var[f(X)]$ is small. This is evidenced by the fact that despite the fact that the two distributions depicted in Figure 3 have similar optimal regression coefficients, $Var[f(X)]$ is two times larger for the distribution depicted on the right. This would help explain the improved performance of ridge-regression, which biases the estimate towards smaller $\lambda$. We explore this hypothesis analytically in Section 6.1.

We next investigate whether these trends are persistent in the research datasets by calculating the ratio of the MSE for PPI++ to the MSE at for Ridge-PPI for each dataset and taking its correlation with $Var[f(X)]$ for that dataset. We find that there is a strong anticorrelation between these values (Pearson r = -0.69), further supporting our hypothesis that small $Var[f(X)]$ hinders the efficacy of PPI++. While we do not have enough datapoints to make this correlation statistically significant, it serves as motivation for the more thorough analysis we present in Section 6.1.

We note that asymptotically, PPI++ is guaranteed to perform at least as well as classical estimation (Angelopoulos et al., 2023b). The failure of PPI++ shown on the left of Fig. 3 can be explained since the sample size is clearly not large enough to reach this asymptotic regime, and so PPI++ is not guaranteed to improve performance. We additionally note that the proof of the PPI++ estimator being asymptotically unbiased presented by Angelopoulos et al. (2023b) does not directly map on to our Sigmoid-PPI approach. While this is

not a concern in the low-label regime, further investigation is necessary to determine the theoretical and empirical efficacy of Sigmoid-PPI when $n$ is large (Appendix A). An interesting path for further work would be determining which statistics of the distribution determine the efficacy of each of these PPI-based approaches.

## 6. Discussion

We now turn our attention to additional analyses to help interpret our results. In particular, we analyze the attributes of the distributions on which PPI++ succeeds, and how our estimation methods may help alleviate problems in scenarios where PPI++ performs poorly. To carry this out, we extend our previous analysis to consider the fact that the $\lambda$ parameter is a random variable; we had previously taken it to be a constant. Our new analysis highlights the role of $Var[f(X)]$ as an important, but previously unconsidered, factor in the success of PPI++.

### 6.1. Stochastic $\lambda$

In this section, we propose a new expression for the variance of PPI++ using the first and second moments of a stochastic $\lambda$ parameter. To simplify these expressions, we assume that $\lambda$ is fit to an independent but identically distributed pool of data. Techniques like cross-fitting have been proposed as a data efficient way to conduct this procedure (Zrnic & Candès, 2024b).

We arrive at the following expression for the excess variance

of PPI, with a full derivation supplied in Appendix C:

$$Var[\hat{\mu}_{PPI}] - Var[\hat{\mu}_h]$$
$$= \mathbb{E}[\lambda]^2 (\frac{1}{N} + \frac{1}{n}) Var[f(X)]$$
$$+ Var[\lambda](2\mathbb{E}[f(X)]^2 + (\frac{1}{N} + \frac{1}{n}) Var[f(X)])$$
$$- \frac{2\mathbb{E}[\lambda]}{n} Cov(h(X), f(X))$$
(11)

This is similar to the expression for the variance of PPI presented in Equation 5 (which assumed a constant $\lambda$), with the notable exception that $Var[\lambda f(X)]$ has been decomposed into two terms depending on the expectation and variance of (stochastic) $\lambda$, respectively. This reveals an important property of PPI++: using a higher variance estimator for $\lambda$ can reduce the efficacy of PPI.

While it is not surprising that the error of a mean estimate will depend on the variance of a variable being used to construct the estimate, this formalism remains important as previous expressions for the variance of PPI neglected $\lambda$-estimation variance as a source of error. This also theoretically justifies the benefits we see from using Ridge-PPI: ridge regression has lower variance than OLS regression, and this expression demonstrates how that may lead to a decrease in estimation error.

### 6.2. Efficacy of PPI++

Further insight into PPI++ specifically can be made if we consider the estimator $\hat{\lambda}_{Opt}$, where

$$\mathbb{E}[\hat{\lambda}_{Opt}] = \frac{\lambda^*}{1 + \frac{n}{N}} = \frac{Cov(f(X), h(X))}{(1 + \frac{n}{N}) Var[f(X)]}.$$
(12)

Furthermore, under certain convergence conditions (see Appendices C and D), $Var[\hat{\lambda}_{Opt}] = \frac{Var[\hat{Cov}[f(X), h(X)]]}{Var[f(X)]^2}$, ultimately giving us:

$$Var[\hat{\mu}_{PPI}] - Var[\hat{\mu}_h]$$
$$= \frac{Var[\hat{Cov}[f(X), h(X)]]}{Var[f(X)]} (\frac{2\mathbb{E}[f(X)]^2}{Var[f(X)]} + (\frac{1}{N} + \frac{1}{n}))$$
$$- \frac{Var[h(X)] Corr(h(X), f(X))^2}{(1 + \frac{n}{N})n}.$$
(13)

Counterintuitively, this demonstrates the variance of PPI++ actually **scales inversely to** $\mathbf{Var[f(X)]}$. This is supported by our empirical findings in Section 5.4. This runs contrary to what one would expect from Equation 5 and further

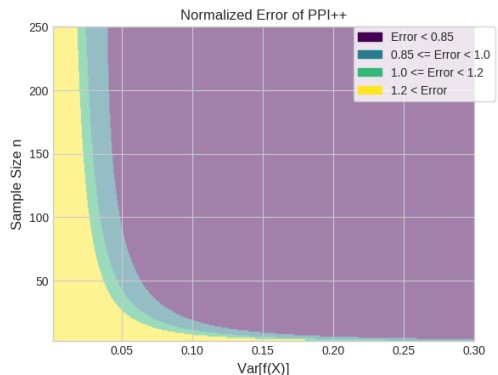

Figure 4: The normalized variance of PPI++, $\frac{Var[\hat{\mu}_{PPI}]}{Var[\hat{\mu}_h]}$, for several values of $Var[f(X)]$ and $n$. $Var[\hat{Cov}(f(X), h(X)]$ is set to a value of 0.3 with an additional $\frac{1}{n^2}$ decay multiplier.

reveals the importance of understanding $\lambda$ as a stochastic quantity.

To end this discussion, we evaluate a number of possible variance values for $\hat{\mu}_{PPI}$, normalized by $Var[\hat{\mu}_h]$, for a distribution from one of our previous experiments. Taking the values of $Corr(h(X), f(X)), Var[h(X)]$ and $\mathbb{E}[f(X)]$ from the Ballots dataset, we plot the variance reduction of PPI++ for various settings of $Var[f(X)]$ and $n$. We take a static value for $Var[\hat{Cov}(f(X), h(X)]$, but we note that the exact relationship between this quantity and $Var[f(X)]$ will likely depend on the data generative procedure which is opaque in most circumstances. Figure 4 demonstrates how a larger $Var[f(X)]$ leads to variance reduction with most smaller values of $n$. Furthermore, it demonstrates that depending on the distributions of $f(X)$ and $h(X)$, specifically if $Var[f(X)]$ is too small, PPI++ will not reduce error in expectation if the labelled sample size is too scarce. This underpins the importance of techniques that reduce the variance of the $\lambda$ estimate, such as Ridge-PPI.

### 6.3. Risk Reduction from Ridge-PPI

To theoretically justify the improvement of Ridge-PPI over PPI++, we can create a similar expression for excess risk as the one presented in Section 6.2. In this section, we refer to the estimator using $\hat{\lambda}_\alpha$ (Equation 8) to calculate $\hat{\mu}_{PPI}$ as $\hat{\mu}_{PPI_\alpha}$. Specifically, we will reason about when Ridge-PPI can improve upon PPI++. Using the same assumptions as Equation 13, as well as the shorthand $V := Var[\hat{Cov}[f(X), h(X)]](2\mathbb{E}[f(X)]^2 + (\frac{1}{N} + \frac{1}{n}) Var[f(X)])$, we show:

$$Var[\hat{\mu}_{PPI_a}] - Var[\hat{\mu}_{PPI}]$$
(14)

$$= V\left(\frac{1}{(Var[f(X)] + \alpha)^2} - \frac{1}{Var[f(X)]^2}\right) \quad (15)$$

$$+ \frac{Corr(f(X), h(X))^2 Var[h(X)]\alpha^2}{n(1 + \frac{n}{N})(Var[f(X)] + \alpha)^2}. \quad (16)$$

A full proof is presented in Appendix E. This expression demonstrates how a small $Var[f(X)]$ contributes to Ridge-PPI improving upon PPI. The first term in Expression 15 presents a variance reduction term, as $V$ is non-negative and the difference is non-positive. When $Var[f(X)]$ is particularly small, $\left(\frac{1}{(Var[f(X)]+\alpha)^2} - \frac{1}{Var[f(X)]^2}\right)$ will be negatively dominated by the latter term in the difference. Importantly, the larger the ridge coefficient $\alpha$, the more negative this difference is, indicating a larger reduction in variance.

Meanwhile, the second term, Expression 16, represents an increase in loss attributable to the fact that Ridge does not yield the optimal $\lambda$. This non-negative term grows as the correlation between $f(X)$ and $h(X)$ increases. This makes intuitive sense, as a stronger correlation between $f(X)$ and $h(X)$ leads to a lower variance estimator when using PPI++. Both these terms in tandem demonstrate how Ridge-PPI can be deployed in circumstances where PPI++ struggles in order to improve performance, namely the circumstance where $Var[f(X)]$ is small.

### 6.4. Optimal Ridge Coefficient

We next look at the optimal selection for the $\alpha$ parameter in Ridge-PPI and use it to further investigate how the statistics of the data relate to PPI's success. To do this, we will apply simple first-order optimization techniques to Expression 14. Note that since this expression is not convex in $\alpha$, it is not guaranteed that this critical point in a minima. Furthermore, depending on the statistics of the data Ridge may not improve upon standard PPI++. However, when the minima is not given by zero or infinity, it will be given by:

$$\alpha^* = \frac{n(1 + \frac{n}{N})V}{Cov(f(X), h(X))^2}. \quad (17)$$

This quantity is difficult to estimate in practice, but supplies more insight into the dynamics of PPI. Notably, this optimum is smaller in the case of greater covariance, and greater in the case of large $V$. Seeing as $V$ is a large positive term in the risk, this expression balances between the potential variance and the potential variance reduction permitted by $Cov[f(X), h(X)]^2$.

## 7. Conclusion and Future Work

In this work, we expanded on the PPI framework to perform mean estimation when very few labelled examples are available. By relating the optimal $\lambda$ setting for PPI++ in mean estimation to the regression coefficient, we provide theoretical motivation for two regression-inspired approaches to mean estimation. Both of these approaches use insights from univariate regression to reduce the variance of the estimate when few labels are available. Through experiments estimating feature generation rates in the context of data analysis and LLM evaluation, we demonstrate that our approaches produce lower variance estimates than both classical estimation as well as PPI in the low-label regime. Through additional analyses and experiments, we further elaborate on when our methods can have the most impact.

The use of predictive models to aid statistical inference is an exciting research frontier. Future work should investigate efficient methods for performing PPI when the labelled and unlabelled samples come from different distributions. Furthermore, the potential impact of the predictive model having varying levels of performance on different subgroups of the distribution and the potential fairness related concerns that come with it is an important future research direction.

## Impact Statement

Our work investigates the shortcomings of the Prediction Powered Inference framework and proposes new methods to help overcome these problems. PPI has been applied to several fields of great impact to society, such as ecological, social science, and biological research, as well as the evaluation of widely used machine learning systems like LLMs. These improvements to PPI have the potential to lead to expedited progress in these application areas by reducing errors in statistical inference. However, while we may have uncovered some cases where PPI does not perform as expected, this is certainly not to be exhaustive. Future work should investigate additional shortcomings of PPI, such as whether bias in the labelled or unlabelled data can lead to fairness concerns.

## Acknowledgements

BE is supported by the funds provided by the National Science Foundation and by DoD OUSD (R&E) under Cooperative Agreement PHY-2229929 (The NSF AI Institute for Artificial and Natural Intelligence).

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

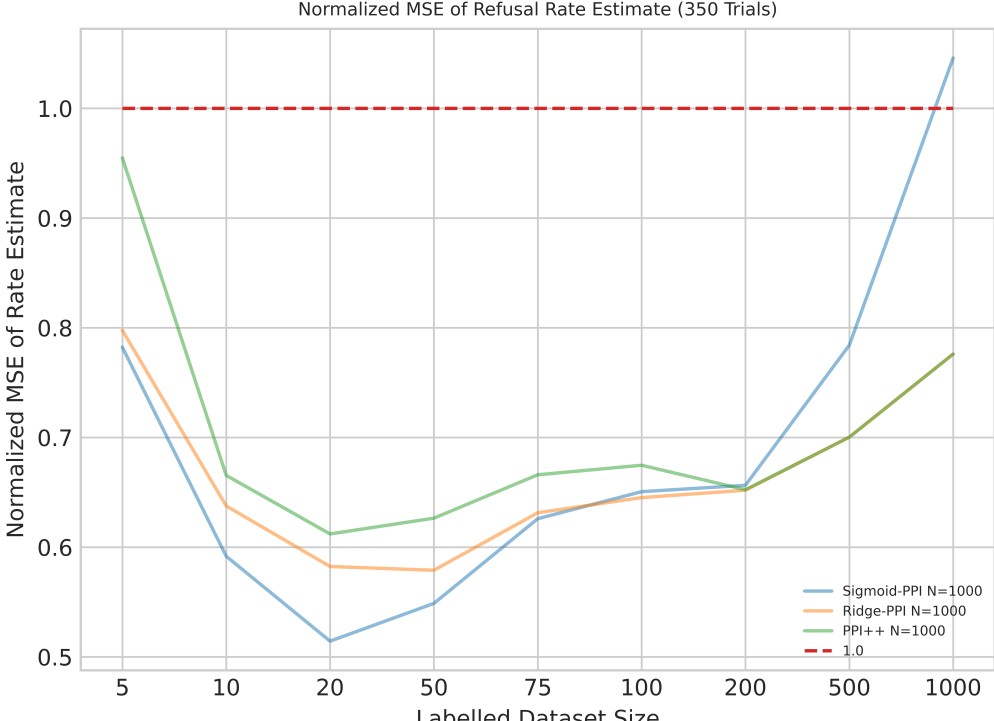

Figure 5: Estimation performance on the LLM refusal dataset with larger labelled batch sizes normalized by the average MAE of classical estimation at that batch size.

## A. Larger Labelled Batch Size Experiments

Though our methods were intended to be used for smaller $n$, we also experiment with the case where both the labelled and unlabelled datasets are large (Figure 5). We find that our methods are best suited for smaller batch sizes, with the greatest improvements over classical estimation (in relative terms) occurring around a labelled batch size of $n = 20$. Beyond this point, we find that the performance of Ridge-PPI converges to the performance of PPI++. This could potentially be explained by the larger pools of labelled data resulting in a smaller $\alpha$ being selected in cross-validation, or simply because the $(1 + \frac{n}{N})$ term in the denominator of our PPI++ estimate $\hat{\lambda}_{Opt}$ and the Ridge-PPI estimate $\hat{\lambda}_\alpha$ (Section 4.4) increases with $n$, causing the estimated $\lambda$ to be closer to 0 and the methods to behave more similarly to classical estimation.

We observe that Sigmoid-PPI's performance decays as the size of the labelled dataset approaches the size of the unlabelled dataset. This could be caused by the lack of a scaling mechanism, such as those described in Section 4.4, that allow the method to rely on classical estimation as $n$ approaches or exceeds the magnitude of $N$, or the presence of a potential asymptotic bias. To counteract this, we propose an adjusted variant of Sigmoid-PPI that also uses a scaling mechanism. Specifically, this estimator has the same form as the one presented in Equation 10, except it uses the following function:

$$g(f(X)) := \frac{1}{1 + \frac{n}{N}} \frac{1}{1 + exp(-\alpha f(X) + \beta)}. \tag{18}$$

Results for this new estimator are presented in Figure 6. We observe that the estimator significantly benefits from this adjustment factor at large labelled dataset sizes: the adjusted estimator continues to make gains over the classical estimate, while the unadjusted estimator may diverge towards excess error.

We recommend using Sigmoid-PPI in cases where the labelled dataset is small, while Ridge-PPI seems to be flexible to different labelled dataset sizes.

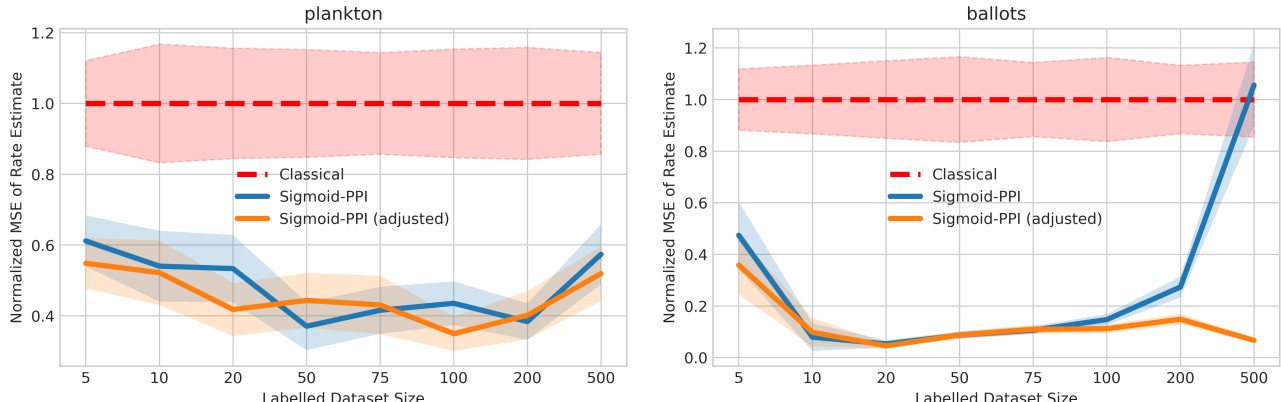

Figure 6: Comparison in performance on the Plankton and Ballots datasets for Sigmoid-PPI and an adjusted Sigmoid-PPI that is scaled by $\frac{1}{1+\frac{n}{N}}$. We see that in some circumstances, the Sigmoid-PPI's normalized error diverges. Meanwhile, the adjustment leads to steady performance gains over the classical baseline.

## B. Proof of Equation 7

In this section, we prove the following expression:

$$\frac{1}{n}\sum_{i=1}^{n}(h(X_i) - \lambda f(X_i) - b)^2 \approx \hat{Var}[h(X)] + \hat{Var}[\lambda f(X)] - 2\hat{Cov}[h(X), \lambda f(X)]. \tag{19}$$

where $b = \hat{\mu}_h - \hat{\mu}_{\lambda f}$ is the optimal ordinary least squares intercept coefficient, which is the difference between the sample means of the target $h(X)$ and transformed input $\lambda f(X)$. We start by expanding the squared loss within the sum.

$$\frac{1}{n}\sum_{i=1}^{n}(h(X_i) - \lambda f(X_i) - b)^2 = \frac{1}{n}\sum_{i=1}^{n}(h(X_i) - \lambda f(X_i) - \hat{\mu}_h + \hat{\mu}_{\lambda f})^2 \tag{20}$$

$$= \frac{1}{n}\sum_{i=1}^{n}((h(X_i) - \hat{\mu}_h) - (\lambda f(X_i) - \hat{\mu}_{\lambda f}))^2 \tag{21}$$

$$= \frac{1}{n}\sum_{i=1}^{n}((h(X_i) - \hat{\mu}_h)^2 + (\lambda f(X_i) - \hat{\mu}_{\lambda f})^2 - 2(\lambda f(X_i) - \hat{\mu}_{\lambda f})(h(X_i) - \hat{\mu}_h)). \tag{22}$$

Distributing the sum and division by $n$ across each of the terms, one can see that each of these terms are biased estimates of $Var[h(X)]$, $Var[\lambda f(X)]$, and $Cov[h(X), \lambda f(X)]$ respectively. Therefore, it's clear that the left hand side of Equation 7 is proportional to the right hand side, and that this expression admits the same minimizing $\lambda$ as the variance depicted in Equation 5.

## C. Stochastic $\lambda$

In this section, we work through the complete decomposition of the variance of PPI assuming an stochastic but independent estimator for $\lambda$.

We analyze the expected mean squared error of PPI++:

$$\mathbb{E}[(\mu_{h_P} - (\frac{1}{N}\sum_{i=1}^{N}\lambda f(X_i^u) + \frac{1}{n}(\sum_{i=1}^{n}h(X_i) - \lambda f(X_i))))^2] = \mathbb{E}[(\mu_{h_P} - (a + \lambda b - \lambda c))^2] \tag{23}$$

To simplify expressions, we take the following shorthands:

$$a = \frac{1}{n}\sum_{i=1}^{n} h(X_i), \quad b = \frac{1}{N}\sum_{i=1}^{N} f(X_i^u), \quad c = \frac{1}{n}\sum_{i=1}^{n} f(X_i) \tag{24}$$

We know that the risk of an MSE loss can be written as a bias-variance decomposition. Since we assume $\lambda$ is independent, this bias is zero. Of course a dependent $\lambda$, which is often the case in practice, would introduce bias.

The variance has the following form:

$$Var[a + \lambda b - \lambda c] \tag{25}$$

$$= Var[a] + Var[\lambda b] + Var[-\lambda c] + 2Cov(a, \lambda b) + 2Cov(a, -\lambda c) + 2Cov(\lambda b, -\lambda c) \tag{26}$$

Since we assume $\lambda$ is independent of our data, we are left with:

$$Var[a + \lambda b - \lambda c] = Var[a] + Var[\lambda b] + Var[-\lambda c] + 2Cov(a, -\lambda c) \tag{27}$$

We will further decompose this expression by analyzing each of its summands:

- **Var[a]**: This is already fully simplified, and is the variance of the classical estimate.
- **Var[$\lambda$b]**: $Var[\lambda b] = Var(\lambda)Var(b) + Var(\lambda)\mathbb{E}[b]^2 + Var(b)\mathbb{E}[\lambda]^2$
- **Var[$-\lambda$c]**: $Var[-\lambda c] = Var(\lambda)Var(-c) + Var(\lambda)\mathbb{E}[-c]^2 + Var(-c)\mathbb{E}[\lambda]^2$
- **2Cov(a, $-\lambda$c)**: $Cov(a, -\lambda c) = \mathbb{E}[-a\lambda c] - \mathbb{E}[a]\mathbb{E}[-\lambda c] = \mathbb{E}[\lambda](\mathbb{E}[-ac] - \mathbb{E}[a]\mathbb{E}[-c]) = \mathbb{E}[\lambda]Cov(a, -c)$

Ultimately this gives us:

$$Var[a + \lambda b - \lambda c] = Var[a] \tag{28}$$

$$+ \mathbb{E}[\lambda]^2(Var(b) + Var(c)) \tag{29}$$

$$+ Var(\lambda)(\mathbb{E}[b]^2 + \mathbb{E}[c]^2 + Var(b) + Var(c)) \tag{30}$$

$$- 2\mathbb{E}[\lambda]Cov(a, c). \tag{31}$$

We can therefore understand the error incurred by a stochastic $\lambda$ as being determined by its variance and expectation. More importantly, this gives us an expression for the excess risk of a stochastic $\lambda$ vs. classical inference:

$$Var[a + \lambda b - \lambda c] - Var[a] \tag{32}$$

$$= \mathbb{E}[\lambda]^2(Var(b) + Var(c)) + Var(\lambda)(\mathbb{E}[b]^2 + \mathbb{E}[c]^2 + Var(b) + Var(c)) - 2\mathbb{E}[\lambda]Cov(a, c) \tag{33}$$

$$= \mathbb{E}[\lambda]^2(\frac{1}{N} + \frac{1}{n})Var(f) + Var(\lambda)(2\mathbb{E}[f]^2 + (\frac{1}{N} + \frac{1}{n})Var(f)) - \frac{2\mathbb{E}[\lambda]}{n}Cov(h, f). \tag{34}$$

Furthermore, if we assume that we are using PPI++ and our $\lambda$ estimator is OLS, we can use the properties that $\hat{\lambda}_{Opt}$, where

$$\mathbb{E}[\hat{\lambda}_{Opt}] = \frac{\lambda^*}{1 + \frac{n}{N}} = \frac{Cov(f(X), h(X))}{(1 + \frac{n}{N})Var[f(X)]}. \tag{35}$$

Furthermore, under certain convergence conditions (see Appendix D), $Var[\hat{\lambda}_{Opt}] = \frac{Var[\hat{Cov}[f(X), h(X)]]}{Var[f(X)]^2}$, ultimately giving us:

$$Var[a + \lambda b - \lambda c] - Var[a] \tag{36}$$

$$= (\frac{Cov(f(X), h(X))}{(1 + \frac{n}{N})Var[f(X)]})(\frac{Cov(f(X), h(X))}{(1 + \frac{n}{N})Var[f(X)]}(\frac{1}{N} + \frac{1}{n})Var(f) - \frac{2}{n}Cov(h, f)) + Var(\lambda)(2\mathbb{E}[f]^2 + (\frac{1}{N} + \frac{1}{n})Var(f)) \tag{37}$$

We will focus on the second factor in the first product:

$$\frac{Cov(f(X), h(X))}{(1 + \frac{n}{N})Var[f(X)]}(\frac{1}{N} + \frac{1}{n})Var(f) - \frac{2}{n}Cov(h, f) \tag{38}$$

$$= Cov(f(X), h(X))(\frac{(\frac{1}{N} + \frac{1}{n})}{(1 + \frac{n}{N})} - \frac{2}{n}) = Cov(f(X), h(X))(\frac{n(\frac{1}{N} + \frac{1}{n}) - 2(1 + \frac{n}{N})}{(1 + \frac{n}{N})n}) \tag{39}$$

$$= Cov(f(X), h(X))(\frac{(1 + \frac{n}{N}) - 2(1 + \frac{n}{N})}{(1 + \frac{n}{N})n}) = Cov(f(X), h(X))(\frac{1 - 2}{n}) = -\frac{Cov(f(X), h(X))}{n} \tag{40}$$

Substituting that back into our original expression, we have:

$$Var[a + \lambda b - \lambda c] - Var[a] \tag{41}$$

$$= (\frac{Cov(f(X), h(X))}{(1 + \frac{n}{N})Var[f(X)]})(-\frac{Cov(f(X), h(X))}{n}) + Var(\lambda)(2\mathbb{E}[f]^2 + (\frac{1}{N} + \frac{1}{n})Var(f)) \tag{42}$$

$$= (-\frac{Cov(f(X), h(X))^2}{n(1 + \frac{n}{N})Var[f(X)]}) + Var(\lambda)(2\mathbb{E}[f]^2 + (\frac{1}{N} + \frac{1}{n})Var(f)) \tag{43}$$

$$= Var(\lambda)(2\mathbb{E}[f]^2 + (\frac{1}{N} + \frac{1}{n})Var(f)) - \frac{Var[h(X)]Corr(h(X), f(X))^2}{(1 + \frac{n}{N})n} \tag{44}$$

Where in the final line we have used the definition of covariance as $Cov(f(X), h(X)) = \sqrt{Var[f(X)]Var[h(X)]}Corr(f(X), h(X))$. Finally, if we substitute in $Var[\hat{\lambda}_{Opt}] = \frac{Var[\hat{Cov}[f(X), h(X)]]}{Var[f(X)]^2}$, we arrive at the expression from Section 6.2:

$$= \frac{Var[\hat{Cov}[f(X), h(X)]]}{Var[f(X)]^2}(2\mathbb{E}[f(X)]^2 + (\frac{1}{N} + \frac{1}{n})Var[f(X)]) - \frac{Var[h(X)]Corr(h(X), f(X))^2}{(1 + \frac{n}{N})n}. \tag{45}$$

# D. Convergence of OLS Estimate

In Section 6.2, we propose that under certain conditions, we can take $Var[\hat{\lambda}_{Opt}] = \frac{Var[\hat{Cov}[f(X),h(X)]]}{Var[f(X)]^2}$. In this section, we set forth the conditions under which that may be the case. Throughout this section we assume we are using the estimator $\frac{\hat{Cov}(f(X),h(X))}{(1+\frac{n}{N})\hat{Var}[f(X)]}$, where $\hat{Var}[f(X)]$ and $\hat{Cov}(f(X),h(X))$ are the sample estimates for the variance of $f(X)$ and covariance of $f(X)$ and $h(X)$, respectively.

In section 6.1, we assumed that the estimator for $\lambda$ used a finite sample separate from the data used for creating the PPI estimate. We will make the additional assumption that we have two pools of data for fitting $\lambda$: one set of $n$ examples for fitting $\hat{Cov}(f(X),h(X))$ and one pool of $N$ examples for fitting $\hat{Var}[f(X)]$, where $n << N$. This has been a standard assumption throughout this work, as we assume that we have access to a very large number of pseudolabels $f(X)$. With that in mind, we rewrite our estimates for these two quantities as $\hat{Cov}_n(f(X),h(X))$ and $\hat{Var}_N[f(X)]$ to denote the random variables being used to generate them, and rewrite our estimator for $\lambda$ as $\frac{\hat{Cov}_n(f(X),h(X))}{(1+\frac{n}{N})\hat{Var}_N[f(X)]}$.

We will now use a set of standard convergence in probability results for the sample variance. Namely, as N grows, we have the following:

$$\hat{Var}_N[f(X)] \longrightarrow_p Var[f(X)], \quad (1+\frac{n}{N})\hat{Var}_N[f(X)] \longrightarrow_p Var[f(X)]. \tag{46}$$

Since this variable converges in probability in a constant, we can use Slutsky's theorem for the following convergence in distribution of our estimator for $\lambda$:

$$\frac{\hat{Cov}_n(f(X),h(X))}{(1+\frac{n}{N})\hat{Var}_N[f(X)]} \longrightarrow_d \frac{\hat{Cov}_n(f(X),h(X))}{Var[f(X)]}. \tag{47}$$

Therefore, in this asymptotic regime, we have:

$$Var[\frac{\hat{Cov}_n(f(X),h(X))}{Var[f(X)]}] = \frac{Var[\hat{Cov}_n[f(X),h(X)]}{Var[f(X)]^2}. \tag{48}$$

Similarly for Ridge-PPI, where $\hat{\lambda}_\alpha := \frac{\hat{Cov}_n[h(X),f(X)]}{(1+\frac{n}{N})(\hat{Var}_N[f(X)]+\alpha)}$ we can use an identical line of reasoning and know that, asymptotically, we have:

$$Var[\hat{\lambda}_\alpha] = Var[\frac{\hat{Cov}_n(f(X),h(X))}{(Var[f(X)]+\alpha)}] = \frac{Var[\hat{Cov}_n[f(X),h(X)]}{(Var[f(X)]+\alpha)^2}. \tag{49}$$

# E. Proof of Excess Risk of Ridge

Here we prove our expression for the difference of risks between Ridge-PPI and the classical estimate:

$$Var[\hat{\mu}_{PPI_a}] - Var[\hat{\mu}_h] \tag{50}$$

Using the decomposition given by Equation 36 in Section C, as well as the fact that $\mathbb{E}[\hat{\lambda}_\alpha] = \frac{Cov(f(X),h(X))}{(1+\frac{n}{N})(Var[f(X)]+\alpha)}$, we can further decompose this expression as:

$$= (\frac{Cov(f(X),h(X))}{(1+\frac{n}{N})(Var[f(X)]+\alpha)})(\frac{Cov(f(X),h(X))}{(1+\frac{n}{N})(Var[f(X)]+\alpha)}(\frac{1}{N}+\frac{1}{n})Var(f) - \frac{2}{n}Cov(h,f)) \tag{51}$$

$$+Var(\lambda)(2\mathbb{E}[f]^2 + (\frac{1}{N}+\frac{1}{n})Var(f)). \tag{52}$$

We'll start by decomposing the first term in the sum. We'll focus on the second factor in the product.

$$\left(\frac{Cov(f(X), h(X))}{(1 + \frac{n}{N})(Var[f(X)] + \alpha)}\left(\frac{1}{N} + \frac{1}{n}\right)Var(f) - \frac{2}{n}Cov(h, f)\right) \tag{53}$$

$$= Cov(f(X), h(X))\left(\frac{(\frac{n}{N} + 1)Var[f(X)] - 2(1 + \frac{n}{N})(var(f(X) + \alpha)}{n(1 + \frac{n}{N})(var(f) + \alpha)}\right) = \frac{Cov(f(X), h(X))}{n}\left(\frac{-Var[f(X)] - 2\alpha}{Var[f(X)] + \alpha}\right) \tag{54}$$

$$= \frac{Cov(f(X), h(X))}{n}\left(-\left(1 + \frac{\alpha}{var(f(X) + \alpha}\right)\right) \tag{55}$$

Substituting this back into the original product, we have:

$$\left(\frac{Cov(f(X), h(X))}{(1 + \frac{n}{N})(Var[f(X)] + \alpha)}\right)\left(\frac{Cov(f(X), h(X))}{(1 + \frac{n}{N})(Var[f(X)] + \alpha)}\left(\frac{1}{N} + \frac{1}{n}\right)Var(f) - \frac{2}{n}Cov(h, f)\right) \tag{56}$$

$$= \left(\frac{Cov(f(X), h(X))}{(1 + \frac{n}{N})(Var[f(X)] + \alpha)}\right)\left(-\frac{Cov(f(X), h(X))}{n} - \frac{\alpha Cov(f(X), h(X))}{(Var[f(X)] + \alpha)n}\right) \tag{57}$$

$$= -\frac{Cov(f(X), h(X))^2}{n(1 + \frac{n}{N})(Var[f(X)] + \alpha)} - \frac{\alpha Cov(f(X), h(X))^2}{n(1 + \frac{n}{N})(Var[f(X)] + \alpha)^2} \tag{58}$$

$$= -\frac{Cov(f(X), h(X))^2 Var[f(X)] + 2Cov(f(X), h(X))^2 \alpha}{n(1 + \frac{n}{N})(Var[f(X)] + \alpha)^2} \tag{59}$$

We now again use the definition of Covariance as $Cov(f(X), h(X)) = Corr(f(X), h(X))\sqrt{Var[f(X)]}\sqrt{Var[h(X)]}$, as well as the fact that $(Var[f(X)] + \alpha)^2 = Var[f(X)]^2 + \alpha^2 + 2Var[f(X)]\alpha$.

$$= -\frac{Corr(f(X), h(X))^2 Var[h(X)](Var[f(X)]^2 + 2Var[f(X)]\alpha)}{n(1 + \frac{n}{N})(Var[f(X)] + \alpha)^2} \tag{60}$$

We'll now add and subtract $Corr(f(X), h(X))^2 Var[h(X)]\alpha^2$ from the numerator:

$$= -\frac{Corr(f(X), h(X))^2 Var[h(X)](Var[f(X)] + \alpha)^2 - Corr(f(X), h(X))^2 Var[h(X)]\alpha^2}{n(1 + \frac{n}{N})(Var[f(X)] + \alpha)^2} \tag{61}$$

$$= \frac{Corr(f(X), h(X))^2 Var[h(X)]\alpha^2}{n(1 + \frac{n}{N})(Var[f(X)] + \alpha)^2} - \frac{Corr(f(X), h(X))^2 Var[h(X)]}{n(1 + \frac{n}{N})}. \tag{62}$$

We now substitute this expression back into the expression for the difference of risks:

$$Var[\hat{\mu}_{PPI_a}] - Var[\hat{\mu}_h] \tag{63}$$

$$= \left(\frac{Corr(f(X), h(X))^2 Var[h(X)]\alpha^2}{n(1 + \frac{n}{N})(Var[f(X)] + \alpha)^2} - \frac{Corr(f(X), h(X))^2 Var[h(X)]}{n(1 + \frac{n}{N})}\right) + Var(\lambda)(2\mathbb{E}[f]^2 + \left(\frac{1}{N} + \frac{1}{n}\right)Var(f)) \tag{64}$$

We can simplify this expression if we make the same simplifying convergence assumptions described in Appendices C and D, as well as the shorthand $V := Var[\hat{Cov}[f(X), h(X)]](2\mathbb{E}[f(X)]^2 + (\frac{1}{N} + \frac{1}{n})Var[f(X)])$:

$$= \frac{V}{(Var[f(X)] + \alpha)^2} + \frac{Corr(f(X), h(X))^2 Var[h(X)]\alpha^2}{n(1 + \frac{n}{N})(Var[f(X)] + \alpha)^2} - \frac{Corr(f(X), h(X))^2 Var[h(X)]}{n(1 + \frac{n}{N})}. \tag{65}$$

Finally, we can also calculate the risk reduction from using Ridge-PPI vs PPI++ by subtracting Equation 13 from Equation 65:

$$Var[\hat{\mu}_{PPI_a}] - Var[\hat{\mu}_h] - (Var[\hat{\mu}_{PPI}] - Var[\hat{\mu}_h]) = Var[\hat{\mu}_{PPI_a}] - Var[\hat{\mu}_{PPI}] \tag{66}$$

$$= \frac{V}{(Var[f(X)] + \alpha)^2} - \frac{V}{Var[f(X)]^2} + \frac{Corr(f(X), h(X))^2 Var[h(X)]\alpha^2}{n(1 + \frac{n}{N})(Var[f(X)] + \alpha)^2} \tag{67}$$

$$= V(\frac{1}{(Var[f(X)] + \alpha)^2} - \frac{1}{Var[f(X)]^2}) + \frac{Corr(f(X), h(X))^2 Var[h(X)]\alpha^2}{n(1 + \frac{n}{N})(Var[f(X)] + \alpha)^2} \tag{68}$$

## F. Optimal Ridge Coefficient

The expression for the risk reduction from using Ridge-PPI vs PPI++ given in Equation 68 is not convex in $\alpha$, and depending on the statistics of the data Ridge may not improve upon standard PPI++. However, if there does exist an optimal setting for $\alpha$, we select it using first order optimization. If we take the derivative of Equation 68 with respect to $\alpha$ we have:

$$\frac{d}{d\alpha}Var[\hat{\mu}_{PPI_a}] - Var[\hat{\mu}_{PPI}] = \frac{2(Corr(f(X), h(X))^2 Var[h(X)]\alpha Var[f(X)] - n(1 + \frac{n}{N})V)}{n(1 + \frac{n}{N})(Var[f(X)] + \alpha)^3}. \tag{69}$$

Setting this to zero we have:

$$\frac{d}{d\alpha}Var[\hat{\mu}_{PPI_a}] - Var[\hat{\mu}_{PPI}] = 0 \implies \alpha = \frac{n(1 + \frac{n}{N})V}{Cov(f(X), h(X))^2} \tag{70}$$

