# OpenReview forum: "Regression for the Mean: Auto-Evaluation and Inference with Few Labels through Post-hoc Regression"
_ICML.cc/2025/Conference — ICML 2025 poster_

### Official Review · Reviewer_UvWk · 2025-03-06

**Overall Recommendation:** 3

**Summary:**

This paper works on statistical inference and its applications in model evaluation. The paper analyzes the pitfalls of a previous method, Prediction Powered Inference (PPI), and proposes two variants to improve performance. The first proposed method is to use ridge regression and add an additional hyperparameter $\alpha$ to the denominator. The other is to use the sigmoid function to improve the expressiveness. Experiments validate the superiority of the proposed method over two baselines.

**Claims And Evidence:**

- The claim that PPI has a high variance when the number of data is limited is true. The proposed method also works well and makes sense. Theoretical and empirical analyses confirm the effectiveness of the proposal.

- I have a concern about what the focus of the paper is. It seems that the proposed method is a more reliable statistical inference method with lower variance. However, it seems that it partially deviates from the model evaluation. The first experiment of benchmark data sets is not related to model evaluation. Therefore, it is more appropriate to say that the paper mainly works on statistical inference rather than model evaluation.

**Essential References Not Discussed:**

There are no essential references that need to be discussed further.

**Experimental Designs Or Analyses:**

- The experiments consist of two parts. The first part is on benchmark datasets used in the related work. The second part is on large language models (LLMs), and the goal is to estimate the true rejection rate. The experiments are extensive and validate the effectiveness in statistical inference. However, since the main purpose of this paper is model evaluation, I think more experiments on automatic evaluation should be included.

- It is appreciated that multiple seeds are run for each experiment. For example, 350 different seeds are considered in the paper. However, only the average is shown in the figures. It is advantageous to consider statistical tests and to show the variances in the figures.

**Methods And Evaluation Criteria:**

The proposed method works well for the problem set. The effectiveness is demonstrated by extensive analysis.

**Other Comments Or Suggestions:**

There are minor typos in the paper. The paper should be checked carefully. For example, there are two "is" in the first sentence of the last paragraph on page 1.

**Other Strengths And Weaknesses:**

Strengths:

- The problem studied in the paper, i.e. more accurate statistical inference, is important and interesting.

- The proposed method is simple but effective.

- Extensive experiments confirm the effectiveness of the proposed method.

- The analysis is good and easy to understand.

Weaknesses:

- It seems that variance reduction is not so rare in the machine learning literature. I wonder if there is related work that also proposes similar ridge regression techniques for variance reduction.

- Since the focus is on model auto-evaluation, the paper should spend more space discussing this and provide more related experiments.

- More detailed descriptions of the proposed method should be included. First, how is the ridge regression problem constructed and how is Eq. (8) derived? Second, why can the introduction of the sigmoid function improve the performance?

**Questions For Authors:**

Please see "Weaknesses".

## update after rebuttal
Thanks for the rebuttal. I will keep my score.

**Relation To Broader Scientific Literature:**

The proposed method is useful for statistical inference and shows promise for application in broader scientific areas.

**Theoretical Claims:**

The theoretical claims are right.

---

> ### Author Rebuttal · Authors · 2025-04-01
>
> We greatly appreciate your review and your suggestions for improving the work. We especially appreciate your thoroughness in reading the paper, and will be sure to fix any typos currently in the draft for the camera ready version of the paper. Below we respond to your individual points.
>
> **Model Evaluation vs. Statistical Inference**
>
> Throughout this work, we empirically and theoretically consider the problem of mean estimation. Statistical inference in the context of science and model evaluation are two interesting applications of this problem, and so we consider them both. However, since these problems are so related, we believe generalizing both of these as mean estimation better represents our methods' broad applicability.
>
> **Error Bars on MSE Plots**
>
> Thank you for your excellent point about the inclusion of error bars in our plots. For the camera ready version of the paper, we will re-run the experiments and include appropriate error bars. Here is a small set of 95\% confidence intervals for the normalized MSE (see figure 1) of different methods across 350 trials. Here the confidence is represented with a tuple (lower bound, upper bound):
>
> |Dataset | PPI (n=5) | Ridge-PPI (n=5)| Sigmoid-PPI (n=5)|
> |--------|------------------|------------------------|--------|
> |Alphafold | (1.12, 1.65) | (0.49, 0.72) | (0.55, 0.82)|
> |Ballots | (0.27, 0.47) | (0.24, 0.44) | (0.34, 0.6)|
> |Census Healthcare | (0.71, 0.95) | (0.68, 0.92) | (0.58, 0.8)|
> |Forest | (1.92, 3.36) | (0.71, 1.22) | (0.82, 1.09)|
> |Galaxies | (0.84, 1.14) | (0.61, 0.9) | (0.71, 0.97)|
> |Plankton | (1.23, 1.89) | (0.56, 0.76) | (0.54, 0.68)|
>
>
> **Sigmoid Function Improving Performance**
>
> Using the sigmoid function in Sigmoid-PPI can improve performance when the mapping from $f(X)$ to $h(X)$ is not linear. For instance, since $h(X)$ is a binary value in the refusal rate dataset, mapping $f(X)$ to (0,1) with a sigmoid function could lead to a lower MSE in the regression problem described in expression 7, which we showed to be equivalent to the variance of PPI.

---

> > ### Comment · Reviewer_UvWk · 2025-04-09
> >
> > Thanks for the rebuttal. I will keep my score.

---

### Official Review · Reviewer_aoGS · 2025-03-13

**Overall Recommendation:** 3

**Summary:**

This paper makes the observation that the variance-minimizing parameter $\lambda$ in PPI++ can be seen (in the scenario where $N$ is large) as a linear regression coefficient when regressing $Y$ on $f(X)$.  Motivated by this observation, this paper proposes an analog of ridge regression, where $\lambda$ is regularized towards zero, and logistic regression, where instead of a linear transformation, a non-linear transformation is used.  Further motivation is given later in the paper, when it is observed that the variance of the PPI++ estimate depends inversely to the variance of $f(X)$.  Experiments on some standard benchmark datasets, as well as a (proprietary?) dataset of LLM refusals demonstrate improved estimation performance (as measured by MAE) compared to PPI++.

**Update after rebuttal**:  As discussed below, I have increased my score from 2 to 3

**Claims And Evidence:**

From a theoretical / methodological perspective, the claims in the paper are somewhat limited and informal, and mainly focused on conceptual goals and intuitions.  From an empirical perspective, the claim of improved estimation seems reasonably well-supported, though I have some questions regarding the setup (see below).

**Essential References Not Discussed:**

(W6) I don't think these missing references are necessarily "essential", but I think they would provide important context as to the novelty of the core observation that $\lambda$ can be interpreted as a regression parameter. While this observation may be a "novel" observation in the context of PPI++, it's hardly novel in the broader context of similar estimators in the statistics community.  [Cheng and Cai 2021], for instance, consider similar linear combinations of estimators in causal inference, and make this connection explicit, see e.g., Equation (9) of [Cheng and Cai 2021] and discussion below it, in particular the discussion of how "the MSE decomposition in (9) has the form of a ridge regression objective". See [Oberst et al. 2022] for a (somewhat out-dated) review of similar estimators that deal with a slightly different setting (since it involves combining biased and unbiased estimators, while in this setting both the classical and PPI++ estimators are unbiased). While the setting is a bit different, those papers consider a problem very closely related to Equation (7) of the present paper, where you want to minimize the MSE of a linear combination of a biased (here, $f(X)$) and unbiased estimator (here, $h(X)$).

[Cheng and Cai 2021] D. Cheng and T. Cai. Adaptive combination of randomized and observational data. arXiv preprint (2111.15012v1), 11 2021. URL: http://arxiv.org/abs/2111.15012v1, arXiv:2111.15012v1.
[Oberst et al. 2022] M. Oberst, A. D' Amour, M. Chen, Y. Wang, D. Sontag, and S. Yadlowsky. Understanding the risks and rewards of combining unbiased and possibly biased estimators, with applications to causal inference. arXiv preprint (2205.10467v2), May 2022. URL: http://arxiv.org/abs/2205.10467v2, arXiv:2205.10467v2.

**Experimental Designs Or Analyses:**

For making the main point, which is that empirical performance (at estimation) improves using the proposed methods.
The experimental design for both the "Research Datasets" (those considered in the original PPI++ paper) and the "LLM Refusal Dataset" seems reasonable enough.

That said, this main analysis could be improved:
* (W1) First, as mentioned in "Methods and Evaluation Criteria", I found the choice of MAE to be somewhat confusing, since the focus otherwise is on reducing variance (and hence, MSE).
* (W3) Second, it would be helpful to include some kind of uncertainty quantification in the lines presented (i.e., the average performance is shown, but not e.g., confidence bounds based on bootstrapping).

(W4) For the secondary analyses, I thought that the empirical investigation could be made a bit more rigorous.  In particular, there is a hypothesis that $\text{var}(f)$ drives performance, and the main evidence provided is somewhat anecdotal (Figure 3).  I would have expected an analysis over the "research datasets" as well, asking e.g., whether it always true that when var(f) is smaller (for some definition of "small"), that the performance of PPI++ is worse relative to classical.  Figure 4 is nice as a simulated example, but it's less clear what the take-away should be in practice.

**Methods And Evaluation Criteria:**

Note: As I will return to some concerns in different sections of the review form, I will number them globally as (W1), (W2), etc, not necessarily in order of importance.

(W1) One of the main weaknesses of this paper, in my view, is the choice of evaluation criteria, along two lines.  First, one of the main goals of PPI++ is providing *valid confidence intervals*, not just more accurate point-estimates as measured by mean-squared error (though these are related concepts), and there is no discussion of that tension here that I could see, nor any discussion on how one would construct confidence intervals from the proposed approach.

(W2) Second, even accepting that the goal of this paper is estimation (rather than inference), the empirical evaluation focuses entirely on mean absolute error, which seems like an odd choice as compared to the more typical mean squared error, especially since the "bias-variance trade-off" being considered is mainly a trade-off seen with squared errors (e.g., the "variance" of an unbiased estimator is exactly the MSE, not the MAE). As a result, I'm not entirely sure what to make of the results in Section 5.

**Other Comments Or Suggestions:**

Typo in the abstract: "PPI++ method can perform even **worse worse** than classical inference"

**Other Strengths And Weaknesses:**

Let's start with strengths: First, as I mention in "relation to the broader scientific literature", I think the goal of this paper is very worthwhile (in particular, giving a critique of PPI++), and I was excited to read it.  The observations, while very conceptual and somewhat heuristic, are nonetheless interesting:  Given that $\lambda$ can be viewed as a regression parameter, why not regularize it, or more explicitly perform regression?

Second, I appreciated the extra intuition provided in Section 6:  If $\lambda$ is independent (easily accomplished via cross-fitting), then its variance plays a role in the excess variance (Equation 11), which gives some intuition and motivation for seeking to minimize it's variance.

Given these strengths, I was a little disappointed not to see these insights get more fully developed, leaving the current paper a bit "thin":
* (W5, see above) On the theoretical side, I would have expected a slightly more rigorous treatment of the impact and motivation for $\alpha$, e.g., what is the impact of $\alpha$ when you push it through the other calculations for e.g., the MSE of the resulting estimator?  The "ridge regression" intuition is a bit hand-wavy, since we typically regularize when there are more parameters than data-points, which is not the case here.
* (W7) On the empirical side, the fact that $\alpha$ is chosen by cross-validation seems to present some additional questions not explored here.  For instance, how does this compare to just directly selecting $\lambda$ using cross-validation?  How robust are these conclusions to the choice of $\alpha$?  What kind of $\alpha$ values end up being selected here?
* (W8) I found the experimental setup to be lacking in some other important details, e.g., on the nature of the LLM refusal dataset (see "Questions for Authors")

**Questions For Authors:**

These questions are directly related to the "weaknesses" I mentioned above, and are given in the order of importance for changing my evaluation.

These questions are most important:
1. (W2) What do the empirical results look like if you use MSE instead of MAE?  Is there a reason to prefer MAE over MSE?
2. (W7) How do these results compare to just directly selecting $\lambda$ using cross-validation?  How robust are these conclusions to the choice of $\alpha$?  What kind of $\alpha$ values end up being selected using the present approach (very large, very small, etc)?
3. (W5) Can you precisely characterize the impact of $\alpha$ on the performance of the estimator, and the theoretically optimal choice of $\alpha$?
4. (W4) What is the association between $\text{var}(f)$ and the performance of PPI++ in the research datasets?  Do the same trends hold?

These questions are less relevant for changing my score, but I'm including them for completeness:
* (W8) Could you say more about the source of the LLM Refusal Dataset, even in anonymous terms?  E.g., is it a proprietary dataset, or is it a public dataset?  How many labelled examples exist to estimate the ground truth refusal rate?  What is the overall refusal rate (e.g., is it very low, or fairly high), etc?

It is not necessary to respond to my weaknesses (W1) and (W6) which I would assume to be handled by adding some language around e.g., broader literature and how this work differs.  (W3) is resolved by just adding some confidence intervals, which I assume is an easy fix that doesn't need to be commented on.

**Relation To Broader Scientific Literature:**

Given the popularity of PPI++ and related methods, the goal of this paper is very worthwhile, and a strength in my view: Investigating failure modes of the method (particularly for smaller sample sizes) and proposing corrections.

(W5) However, the observation that the $\lambda$ parameter in PPI++ is related to a regression coefficient is a relevant observation, but it's not entirely novel or surprising, as I discuss under "Essential References".  The findings of this paper are primarily empirical: While there is some connection to ridge regression, there isn't a very rigorous theoretical argument presented as to why "shrinkage" should be beneficial here, other than the loose intuition that it might reduces the variance of $\lambda$.

**Theoretical Claims:**

There isn't much in terms of theoretical claims in this paper, but the claims as written appear to be correct.  There are some opportunities to improve the clarity and precision with which the theoretical claims are made, which I address under "Other Strengths and Weaknesses"

---

> ### Author Rebuttal · Authors · 2025-04-01
>
> Thank you very much for your insightful review. Your questions and comments have helped to make the draft much stronger and clearer. In the camera ready version of the paper, we will improve the related works section and better describe the LLM refusal dataset. We address your individual concerns below.
>
> **Use of MAE instead of MSE**
>
> Evaluating our method in terms of MSE is an excellent suggestion. MAE was initially used as it is more interpretable. In the camera-ready version of the manuscript, we will be sure to use MSE instead of MAE.
>
> This change produces nearly identical results to that which we presented in the paper. We have included a small set of results to demonstrate this.
>
>
> |Dataset | MSE of PPI (n=5) | MSE of Ridge-PPI (n=5)|
> |--|---|---|
> |Alphafold | 0.042 | 0.019 |
> |Ballots | 0.391 | 0.307 |
> |Census Healthcare | 0.445 | 0.356 |
> |Forest | 0.034 | 0.015 |
> |Galaxies | 0.075 | 0.041 |
> |Plankton | 0.03 | 0.012 |
>
>
>
> **Expanding Section 6, Further Theoretical Justification for Ridge-PPI**
>
>
> Several reviewers requested further investigation of the theoretical analyses on where PPI++ succeeds and fails in section 6, as well as an extended theoretical justification of Ridge-PPI.
>
> Towards addressing these concerns, we below produce an expression for the excess risk of Ridge-PPI - an analogous expression to Expression 13 in the paper that makes the same convergence and independence assumptions. We use the shorthand $\hat{\mu}_{PPI_a}$ for the PPI estimate (Expression 3 in the paper) using the Ridge-PPI lambda (Expression 8), as well as $V:= Var\[\hat{Cov}[f(X), h(X)]\](2\mathbb{E}[f(X)]^2 + (\frac{1}{N} + \frac{1}{n})Var[f(X)])$:
>
>
> \begin{equation}
>     Var[\hat{\mu}_{PPI_a}] -Var[\hat{\mu}_h]
>     = \frac{V}{(Var[f(X)] + \alpha)^2}
> \end{equation}
>
> \begin{equation}
>     - \frac{Var[h(X)]Corr(h(X),f(X))^2}{(1+\frac{n}{N})n} + \frac{Var[h(X)]Corr(h(X),f(X))^2\alpha^2}{(1+\frac{n}{N})n(Var[f(X)] + \alpha)^2}
> \end{equation}
>
> Taking the derivative of this expression and setting it to zero yields the solution:
>
> \begin{equation}
>     \alpha^* = \frac{(1+\frac{n}{N})nV}{Cov[f(X), h(X)]^2}
> \end{equation}
>
> This quantity is difficult to estimate in practice, but supplies more insight into the dynamics of PPI. Notably, this optimal $\alpha$ is smaller in the case of greater covariance, and greater in the case of large V. Seeing as V is a large positive term in the risk, this expression balances between the potential variance and the potential variance reduction permitted by $Cov[f(X), h(X)]^2$.
>
> We can also now calculate the expected risk reduction from using Ridge-PPI vs PPI++:
>
> \begin{equation}
>    Var\[\hat{\mu}_{PPI_a}\] - Var\[\hat{\mu}PPI\]
> \end{equation}
>
>
> \begin{equation}
>     = \frac{\alpha}{(Var[f(X)] + \alpha)^2}(\frac{Var[h(X)]Corr(h(X),f(X))^2\alpha}{(1+\frac{n}{N})n} - V\frac{2Var[f(X)] + \alpha}{Var[f(X)]^2})
> \end{equation}
>
> This expression further demonstrates the potential harms of small $Var[f(X)]$: the smaller $Var[f(X)]$ is, the greater the improvement from using Ridge-PPI over PPI++.
>
> **Cross Validating $\lambda$**
>
> We performed some initial experiments using cross validation to fit a scalar regression parameter on a synthetic regression task. We found that while it can perform well in some circumstances, it requires two key assumptions: 1) that one knows a priori the possible range of regressor values, and 2) that one can use a dense grid of values to validate over (e.g. a grid of 50 options in the [0, 1] interval works reasoanbly well, a grid of 10 does not).
>
> When these are both true, this cross-validated regression strategy is a constrained function class that can reduce variance. However, our use of cross validation for selecting an $\alpha$ parameter did not have these assumptions, as the $\lambda$ values we were effectively searching over were exclusively ridge regressors. We appreciate this comment, however, as it helped us to improve our understanding of the scalar regression problem and exemplified the computational efficiency of Ridge-PPI.
>
> **Creating Confidence Intervals**
>
> Reviewer NpF2 shared this concern. Due to the lack of a general rebuttal as well as space restrictions on individual rebuttals, we have included the response to this in the rebuttal to reviewer NpF2. Please read it to see our additional explanation for this issue.
>
> **Research Datasets and Var[f(X)]**
>
> We investigate whether the trends observed in Section 5.4 are persistent in the research datasets by calculating the ratio of the MSE at $n=5$ for PPI++ to the MSE at $n=5$ for Ridge-PPI for each dataset and taking its correlation with $Var[f(X)]$ for that dataset. We find that there is a strong anticorrelation between these values (Pearson r = -0.69), further supporting our hypothesis that small $Var[f(X)]$ hinders the efficacy of PPI++. While we do not have enough datapoints to make this correlation statistically significant, it serves as motivation for the more thorough analysis we present in section 6.

---

> > ### Comment · Reviewer_aoGS · 2025-04-07
> >
> > I appreciate the additional derivations provided, and I'm glad to see that the story doesn't change much when considering MSE vs. MAE.  I would encourage the authors to include these derivations in the final version, as well as others discussed in other responses (e.g., the calculation of the bias when $\lambda$ is not fit independently).
> >
> > While I have not had time to verify the calculations myself, and while I find that many of these contributions are a bit incremental individually, I think they are substantial enough together to warrant increasing my score.

---

> > > ### Author Response · Authors · 2025-04-07
> > >
> > > Thank you very much for your thoughtful response to our rebuttal. We will be sure to include these amendments in the camera-ready version of the paper.

---

### Official Review · Reviewer_NpF2 · 2025-03-13

**Overall Recommendation:** 2

**Summary:**

The paper focuses on the specific scenario where PPI is used for mean estimation. The authors note an equivalence between the tuning parameter estimation for PPI++ and linear regression, and examine several modifications that can be derived. Results are validated through empirical evidence on several benchmark data sets.

**Claims And Evidence:**

The paper claims to provide a theoretical analysis of the link between PPI++ and linear regression, as well as building extentions for PPI derived from this link. Both of those claims are well supported.

While the paper clearly states it is focused on mean estimation (it's in the title of the paper!), the abstract makes no reference to this and could be misleading. I would recommend the abstract is amended to be consistent.

**Essential References Not Discussed:**

All necessary references appeared to be included

**Experimental Designs Or Analyses:**

Experimental design was sound.

Regarding section 5.4, I do not clearly understand what this evaluation is doing. I understand the input dataset comes from multiple LLMs, and my understanding is the dataset is being partitioned by LLM for this section. Given the following Section 6 is an extended discussion of these results, more framing may be appropriate. In particular, make it clear up front that the different LLMs have different variances, and perhaps discuss why this is true, if the reason is known.

**Methods And Evaluation Criteria:**

While the link between PPI++ and linear regression is explored theoretically, the extensions from sections 4.2 and 4.3 are only evaluated empirically. Given the concerning empirical evaluation in Appendix A, more theoretical evaluation would be preferable.

**Other Comments Or Suggestions:**

- Should equation 9 be $g(f(X)) := \frac{1}{1+exp(-af(X) + b)}$?
- The abstract contains the word 'worse' repeated: "can perform even worse worse than classical inference"
- I'm not sure this paper is an appropriate place to call out a biased implementation of software.

**Other Strengths And Weaknesses:**

The paper proposes several straightforward extensions to PPI for mean estimation that will likely improve results in practice. The exposition is generally very clear and easy to follow. The discussion in Section 6 is quite intriguing.

However, while the paper presents several intriguing ideas, it ultimately feels somewhat incomplete. For example:
- The paper only focuses on mean estimation, and makes no attempt to cover more general M-estimation that the original PPI papers cover.
- There is no discussion or empirical results for inference procedures such as confidence intervals.
- Footnote 1 mentions the introduction of bias when $n$ is small, but there is no further theoretical or empirical evaluation of this bias.
- The results of Appendix A are concerning, and potentially undermine the extension in Section 4.3.
- The discussion in Section 6 is very interesting, but feels incomplete.

**Questions For Authors:**

1. Is this method relevant for more general M-estimation? The equivalence to OLS is obvious for mean estimation, but perhaps something analogous exists for other estimates of interest?
2. Do the proposed extensions impact confidence intervals while retaining coverage, at least empirically?
3. Following footnote 1, have you performed an empirical evaluation of whether the bias exists in the empirical results?

**Relation To Broader Scientific Literature:**

This paper discusses potential extensions for PPI, which is a general procedure for statistical inference that could impact many fields. That said, the extension is focused on one particular use case (mean estimation), which arguably is the most important use case, but nonetheless could limit the impact of the results. The discussion in Section 6 could be critical for the PPI literature as a whole, and deserves further study.

**Theoretical Claims:**

Proofs and derivations appeared correct.

---

> ### Author Rebuttal · Authors · 2025-04-01
>
> We would like to express our appreciation for your thoughtful review and constructive critique. We look forward to making the writing more clear following your points. Specifically, we will amend the abstract and make the description of the LLM refusal dataset more clear. We respond to individual points of yours below.
>
> **Appendix A**
>
> Sigmoid PPI is intended for use with small sets of labelled examples. Within this regime, we see Sigmoid PPI regularly match the original PPI++ or make significant improvements.
>
> However, one potential fix for the results seen in Appendix A would be using the following alternative formula:
>
> \begin{equation}
>     g'(f(X)) := \frac{1}{1 + \frac{n}{N}}\frac{1}{1 + exp(-\alpha f(X) + \beta)},
> \end{equation}
>
> This would have the estimator behave more similarly to the classical estimator as n -> N and the classical estimator becomes stronger. We are currently experimenting with this modification and hope to include them in the camera ready version of the paper. Below we provide promising results of average MSE at the largest possible $n$ for the research datasets, demonstrating how this method (Sigmoid-Div) improves asymptotic performance:
>
> |Dataset | PPI | Sigmoid-PPI | Sigmoid-Div|
> |--------|------------------|------------------------|--------|
> |Alphafold | 8.94e-05 | 0.0001 | 7.03e-05|
> |Ballots | 1.6e-05 | 0.0002 | 1.53e-05|
> |Forest | 0.00013 | 0.00018 | 0.00015 |
> |Galaxies | 0.00024 | 0.00023 | 0.0002|
> |Plankton | 0.00014 | 0.00013 | 0.00012|
>
>
> **Expanding Section 6, Further Theoretical Justification for Ridge-PPI**
>
> This was a shared concern for reviewer aoGS as well. Due to the lack of a general rebuttal as well as space restrictions on individual rebuttals, we have included the response to this in the rebuttal to reviewer aoGS. Please read it to see our additional analysis for this issue.
>
> **Bias of PPI++**
>
> Footnote 1 in the paper indicates that for small $n$, PPI++ will be a biased estimator.
> Prior work only considered the asymptotic unbiasedness of $\lambda$ and there is no existing expression for this bias with small samples.
> Here, we produce such an expression and are happy to include it in the paper if the reviewer finds  it helpful.
> The expression for this bias is the following (revealed by taking the expectation of PPI assuming $\lambda$ is not independent from the training data):
>
> \begin{equation}
>     E[\hat{\mu}_{PPI}] - \mu^* = -Cov[\hat{\lambda}, \hat{\mu}_f]
> \end{equation}
>
> We note that this expression can be applied to any estimator of $\lambda$, such as Ridge-PPI. Using the definition of covariance can make this expression more interpretable:
>
> \begin{equation}
>     E[\hat{\mu}_{PPI}] - \mu^* = \sqrt{Var[\hat{\lambda}]Var[\hat{\mu}_f]}Corr[\hat{\lambda}, \hat{\mu}_f]
> \end{equation}
>
> We note that we generally do not include this bias term as it can be ignored in cases where we assume that the $\lambda$ parameter can be fit independently, such as section 6.1. Furthermore, since our methods focus on variance reduction (in line with prior literature) and were effective at reducing overall MSE/MAE, we consider a complete treatment of this bias to be out of scope for this work.
>
> **Other M-Estimators**
>
> We chose to focus our work on specific insights into the dynamics of PPI when used with mean estimation.
> While other M-estimators are interesting as well, we find them out of scope for our work, which is specifically about the small sample behavior of PPI for mean estimation. We believe the small sample focus of our work validates the contribution as novel and useful, despite not considering a wide range of M-estimators.
>
> **Creating Confidence Intervals**
>
> We note that throughout this work we did not consider the efficacy of constructing confidence intervals for any regime, since the central limit theorem (which past methods rely on) tends not to hold in small data regimes. Since improving the efficacy of PPI in the small data regime was our primary concern, those intervals would not be valid for any of the methods that we compared with.
>
> However, the basic machinery from Angelopoulos et al. [2023] will work to create confidence intervals for Ridge-PPI, as constructing the interval is not impacted by how $\lambda$ is selected.
> Additionally, for Sigmoid-PPI, assuming that the parameters of the transformation are fit independently as in section 6.1, the confidence interval can be created using a very similar formula to Angelopoulos et al. [2023], with the exception that the transformed predictions are used in place of the original predictions:
>
> \begin{equation}
>     \hat{\mu}_{PPI_g} \pm 1.96*\sqrt{\frac{\hat{Var}[g(f(X_i)) - h(X_i)]}{n} + \frac{\hat{Var}[g(f(X_i^u))]}{N}}
> \end{equation}

---

### Official Review · Reviewer_V1XE · 2025-03-17

**Overall Recommendation:** 4

**Summary:**

The paper provides a method to improve over standard prediction-powered inference by weighting the correction factor in the same way that is done in control variates. The methods are simple and improve over existing baselines, and the authors provide an illuminating discussion.

**Claims And Evidence:**

Claims were clear and method is clear.

My only concern is that of the variance of the estimated lambda that occurs due to estimation on data. For example, it would have been very useful to see the derivation with the variance with the optimal lambda_hat plugged in. Something like this was attempted in 6.2, but more the estimated covariance term still exists in there, which makes the math less illuminating than seems possible.

**Essential References Not Discussed:**

None here.

**Experimental Designs Or Analyses:**

Experiments are valid and support the new methods clearly.

**Methods And Evaluation Criteria:**

Methods are clear, evaluation criteria are sound.

The main idea for the method is to see that lambda can be seen as a scalar "corrector" for f(X) to approximate h(X), whose mean one wants to estimate. This whole idea can more generally be thought of as a special case of variance reduction by averaging a conditional mean, and the conditional mean is estimated by regressing h(X) on f(X) over a. simple function class that can be fit on few samples. The papers uses linear and logistic regression.

The evaluation of the method is done on estimating means from few samples in problems ranging from protein function to segmentation.
The evaluation showed that on all the small sample sizes considered, the proposed modifications to PPI improves over vanilla mean estimation, and also other modifications in the literature.

I like the refusal rate example.

Also, the authors should provide confidence intervals for the experiments.

**Other Comments Or Suggestions:**

Please improve the quality of the figures, with larger font sizes and HD figures.

**Other Strengths And Weaknesses:**

Simple method, clear new insight about how to interpret PPI, and it works.

**Questions For Authors:**

-

**Relation To Broader Scientific Literature:**

The method, as it stands, improves on the PPI methodology and the related work discusses relavant work about PPI modifications.

**Theoretical Claims:**

Theory is correct.

---

> ### Author Rebuttal · Authors · 2025-04-01
>
> First of all, we thank you for your constructive feedback and positive reception of the work. We look forward to improving the manuscript based on your review. We will be sure to improve readability of the figures for the camera ready draft. Below, we respond to individual points you raised.
>
> **Optimal Lambda Risk Expression**
>
> While Expression 5 in the paper does describe the risk of PPI for a static lambda, we substitute the optimal lambda presented in Expression 6 below.
>
> \begin{equation}
>     =  \frac{1}{n}(Var[ h(X) ] + \frac{1}{Var[f(X)]} -  2 Var[h(X)]Corr[h(X), f(X)]^2).
> \end{equation}
>
> **Error Bars on MSE Plots**
>
> Thank you for your excellent point about the inclusion of error bars in our plots. For the camera ready version of the paper, we will re-run the experiments and include appropriate error bars. Here is a small set of 95\% confidence intervals for the normalized MSE (see figure 1) of different methods across 350 trials. Here the confidence is represented with a tuple (lower bound, upper bound):
>
> |Dataset | PPI (n=5) | Ridge-PPI (n=5)| Sigmoid-PPI (n=5)|
> |--------|------------------|------------------------|--------|
> |Alphafold | (1.12, 1.65) | (0.49, 0.72) | (0.55, 0.82)|
> |Ballots | (0.27, 0.47) | (0.24, 0.44) | (0.34, 0.6)|
> |Census Healthcare | (0.71, 0.95) | (0.68, 0.92) | (0.58, 0.8)|
> |Forest | (1.92, 3.36) | (0.71, 1.22) | (0.82, 1.09)|
> |Galaxies | (0.84, 1.14) | (0.61, 0.9) | (0.71, 0.97)|
> |Plankton | (1.23, 1.89) | (0.56, 0.76) | (0.54, 0.68)|

---

### Decision · Program_Chairs · 2025-05-01

**Decision:**

Accept (poster)

**Comment:**

This paper proposes a simple extension of the PPI methodology via an interpretation as post-processing a biased estimate. While straightforward, the method seems to work well. The reviewers appreciated the contributions; especially the strength of the empirical results. However, they also raised a number of concerns (on clarity, details for the alpha hyperparameter, MSE instead of MAE, confidence intervals,  etc),  which should be addressed for the final version.

Some math/style issues: use \widehat instead of \hat for better style (e.g., \widehad Cov).